# Distribution of psychrophilic microorganisms in a beef slaughterhouse in Japan after cleaning

**Ayaka Nakamura**[1], **Hajime Takahashi**[1]*, **Anrin Kondo**[1], **Fumiaki Koike**[2], **Takashi Kuda**[1], **Bon Kimura**[1], **Mitsushi Kobayashi**[2]

**1** Department of Food Science and Technology, Tokyo University of Marine Science and Technology, Tokyo, Japan, **2** Hida Meat Agricultural Cooperative Association, Takayama City, Gifu, Japan

* hajime@kaiyodai.ac.jp

**Data Availability Statement:** All relevant data are within the manuscript and its Supporting information files.

## Abstract

The purpose of this study was to investigate the abundance and distribution of psychrophilic microorganisms associated with spoilage in beef slaughterhouse environments after cleaning. The processing lines and equipment used in slaughtering and boning were swabbed, and the microbial count was determined using a TSA and MRS medium and Chromocult® Coliform agar incubated at 15°C and 37°C, respectively. As a result, the brisket saw (handle side) and trolley hook were the most heavily contaminated with microorganisms, with each having a microbial adhesion rate of 66.7%. The microbial adhesion rates of the apron and milling cutter (edge side) were 50%, respectively, and those of the foot cutter (edge and handle side), splitting saw (edge side), and knife (handle side) were 33.3%, respectively. Next, four colonies were randomly isolated from the petri dish used for the bacterial count measurement to identify the predominant microbial species of the microorganisms attached to each equipment. As a result of Sanger sequencing analysis, yeasts such as *Candida zeylanoides* and *Rhodotorula* sp. and bacteria including *Pseudomonas* sp. and *Rhodococcus* sp. were identified from the equipment used in the slaughtering line, and it was assumed that these microorganisms were of environmental origin. In contrast, only *Pseudomonas* sp. and *Candida zeylanoides* were isolated from the boning line. Despite the use of cleaning operations, this study identified some equipment was contaminated with microorganisms. Since this equipment frequently comes into direct contact with the carcass, it is critical to thoroughly remove the microorganisms through accurate cleaning to prevent the spread of microbial contamination on the carcasses.

## Introduction

Beef carcasses become contaminated with microorganisms during the slaughter and subsequent deboning processes. Many pathogenic bacteria are present in cattle skin and intestinal contents, and the contamination spreads by adhering to the carcass surface after skinning [1–3]. Since the slaughterhouse is the furthest upstream from food distribution, strict hygiene

**Funding:** The author(s) received no specific funding for this work.

**Competing interests:** The authors have declared that no competing interests exist.

management is required to prevent the spread of harmful bacteria downstream. Process control by the Hazard Analysis and Critical Control Point (HACCP) system is one measure to prevent the contamination of carcasses by such harmful bacteria. In Japan, following the EU and the United States, introducing the HACCP system in slaughterhouses became mandatory from June 2021 [4]. In addition to the HACCP system, compliance with Good Manufacturing Practice (GMP) is essential. Careful hygiene management, such as maintenance and inspection of equipment and machinery, is required to produce safe meat.

Bacteria associated with food poisoning, such as enterohemorrhagic *Escherichia coli* and *Salmonella* are among the hazardous bacteria subject to critical control point (CCP) management in cattle slaughterhouses [5]. These food-poisoning bacteria are mesophilic facultative anaerobes in the cattle's intestinal tract. When feces or intestinal contents adhere to the carcass, trimming to completely remove the contaminated section effectively controls pathogenic bacteria. Furthermore, the chilling process after processing the carcass is recognized as CCP to prevent the growth of harmful bacteria by carefully controlling the low temperature [6].

In recent years, the international trading volume of beef has been increasing annually, and beef with a long expiration date has become advantageous in terms of sales strategy. The international beef trade is transported by refrigeration and freezing [7], but transportation under refrigerated conditions is preferable for parts with high unit prices, such as steak meat, because freezing conditions degrade beef quality [8]. However, long transportation periods can allow some low-temperature-growing microorganisms to grow and degrade beef quality [9, 10]. The number of bacteria that attach to the beef at the initial stage is the most important factor in producing beef that can withstand long-term transportation in a refrigerated state. If the number of initial bacteria in beef can be reduced as much as possible, the period until spoilage can be extended.

After skinning, the surface of beef carcasses has a low bacterial count, but microbial contamination accumulates through the equipment, lines, employee gloves, and aprons used in the subsequent process [11, 12]. In slaughterhouses with good hygiene management, the general bacterial count is investigated as part of the hygiene inspection after cleaning the equipment and lines. However, as described above, these are frequently targeted at medium-temperature bacteria that can be pathogenic, and psychrophilic microorganisms that cause spoilage during refrigerated distribution have received little attention. It is critical to prevent psychrophilic microorganisms from adhering to the carcass to produce beef with a long-term shelf life. However, to the best of our knowledge, no studies have been conducted on the distribution of psychrophilic microorganisms in slaughterhouse equipment and lines. The purpose of this study was to understand the actual contamination of psychrophilic microorganisms in the equipment used in the slaughtering line and the boning line.

## Materials and methods

### Swab sampling

Sampling was carried out at the slaughterhouse of the Federation of Hida Meat Agricultural Cooperative Association. The equipment was tested after the cleaning operation. At this facility, a boning line is attached to the slaughtering line, and a series of tasks are performed from the slaughter line to process the body into carcasses, followed by cutting and packaging at the boning line. Cleaning was carried out after each line was used, and the equipment was wiped the day after cleaning. Between 2019 and 2021, a total of six samplings on the slaughter line (Trial 1–6; 7/31/2019, 8/7/2019, 7/29/2020, 8/5/2020, 1/7/2021, 1/13/2021), and seven samplings on the boning line (Trial 1–7; 8/7/2019, 12/12/2019, 12/18/2019, 7/29/2020, 8/5/2020, 1/7/2021, 1/13/2021) were carried out. The knife, foot cutter, brisket saw, splitting saw, milling

cutter, dehider, trolley hook, and aprons used in the slaughtering line were wiped off (S1 Fig). In the boning line, the large division conveyor belt, boning conveyor belt, turntable, meat holder, and electric saw were swabbed (S2 Fig). The saws were sampled twice: once on the edge side and once on the handle side. For the wiping inspection, a commercially available swab (Elmex, Tokyo, Japan) was used, and the entire surface was swabbed for those with complicated shapes, and 100 cm$^2$ was swabbed for equipment with flat surfaces such as aprons, large division conveyor belt, boning conveyor belt, and the turntable.

## Measurement of bacterial counts

The number of bacteria in each swab sample was measured on three types of media: trypticase soy agar (TSA; Becton, Dickinson and Company, Franklin Lakes, NJ, USA) for general bacteria, de Man, Rogosa, and Sharpe (MRS) agar (Merck KGaA, Darmstadt, Germany) for lactic acid bacteria, and Chromocult® Coliform Agar (Merck) for coliform bacteria and *E. coli*. The swab sample was diluted ten-fold with saline solution, and 100 μL of each original solution and diluent were spread onto each agar medium. TSA and MRS media were incubated at 15°C for 96 h, while Chromocult® Coliform Agar was incubated at 37°C for 24 h. After incubation, the number of colonies were counted. The number of bacteria per 1 cm$^2$ was calculated for aprons, the large division conveyor belt, the boning conveyor belt, and the turntable, while the number of bacteria per 1 mL of the swab was calculated for other equipment with complicated shapes. All experiments were duplicated, and those with an average number of colonies of 5 were regarded as the detection limit for reliability (5 CFU / cm$^2$ or $5.0 \times 10^1$ CFU/swab).

## Isolation of microorganisms

Four colonies were selected from TSA medium or MRS medium, and the number of colonies that exceeded the detection limit was confirmed. The bacterial species were identified through sequencing analysis. Since the selected colonies may be bacterial or yeast, morphological observations were made prior to DNA extraction with a optical microscope. DNA extraction was carried out under optimal conditions for yeast and bacteria, as described below. Bacterial isolates from TSA and MRS media were cultured at 15°C for 48 h in trypticase soy broth (TSB) (Becton, Dickinson and Company) and MRS broth (Merck KGaA), respectively. The yeast isolates were cultured at 15°C for 48 h in yeast peptone dextrose (YPD) broth, which was prepared with 20 g/L peptone (Becton, Dickinson and Company), 10 g/L yeast extract (Becton, Dickinson and Company), and 20 g/L glucose (Kokusan Chemical Co., Ltd., Tokyo, Japan). After culturing, 1 mL of the enriched bacterial solution was centrifuged at $15,000 \times g$ for 3 min.

## Identification of isolated microorganisms

The supernatant was removed to obtain pellets for DNA extraction. Bacterial DNA extraction was performed using Nucleospin (Macherey-Nagel, Düren, Germany) according to the manufacturer's instructions. After DNA extraction, the 16S rRNA region of the bacteria and the D1 / D2 large-subunit (LSU rRNA region of yeast were amplified using a GeneAmp 9700 Thermal Cycler) (Life Technologies, Carlsbad, CA, USA). For the 16S rRNA region and D1 / D2 region, universal primers 27F (5′-AGA GTT TGA TCC TGG CTC AG-3′) and 1492R (5′-GGT TAC CTT GTT ACG ACT T-3′), NL1 (5′-GCA TAT CAA TAA GCG GAG GAA AAG -3′), and NL4 (5′-GGT CCG TGT TTC AAG ACG G-3′) were used, respectively. The polymerase chain reaction (PCR) conditions for bacteria were as follows: initial denaturation at 94°C for 4 min, 30 cycles of amplification (94°C for 30 s, 58°C for 1 min, and 72°C for 1 min), and final extension at 72°C for 4 min. The PCR conditions for yeast were as follows: initial

denaturation at 94˚C for 5 min, 40 cycles of amplification (94˚C for 30 s, 51˚C for 1 min, and 72˚C for 5 min), and final extension at 72˚C for 5 min. The PCR products were purified using AM Pure XP (Beckman Coulter, Brea, CA, USA) and sent to Eurofins Genomics (Eurofins Genomics, Tokyo, Japan) for sequencing with the 27F and NL1 primers. The basic local alignment search tool (BLAST) algorithm was used to compare the derived sequences with the 16S rDNA sequences or 26S rRNA sequences in the DNA Data Bank of Japan database (http://blast.ddbj.nig.ac.jp/blastn, Shizuoka, Japan).

# Result and discussion

## Distribution of psychrophilic microorganisms in equipment and lines

**Slaughtering line.** Despite being swabbed after cleaning, all equipment used in the slaughtering line, except for the dehider (edge side), Brisket saw (edge side), and milling cutter (handle side), were found to have microbial adherence in at least one trial (Table 1). *Escherichia coli* and coliform bacteria, both of which are mesophilic bacteria, were not detected in any of the trials from the equipment used in the slaughter and boning lines. Among the equipment

**Table 1. The number of psychrophilic microorganisms adhering to equipment used in slaughtering lines.**

| Zone where equipment is used | Sampling point | The side that was wiped off | Bacterial counts in each medium (log CFU/ml) ** | | | | | | | | | | | | Positive rate (%) |
|---|---|---|---|---|---|---|---|---|---|---|---|---|---|---|---|
| | | | Trial-1 | | Trial-2 | | Trial-3 | | Trial-4 | | Trial-5 | | Trial-6 | | |
| | | | 2019/7/31 | | 2019/8/7 | | 2020/7/29 | | 2020/8/5 | | 2021/1/7 | | 2021/1/13 | | |
| | | | TSA | MRS | TSA | MRS | TSA | MRS | TSA | MRS | TSA | MRS | TSA | MRS | |
| Dirty zone | Foot cutter | Edge side | N.D. | N.D. | 2.9 | N.D. | N.D. | N.D. | N.D. | N.D. | 1.4 | 1.7 | N.D. | N.D. | 33.3 |
| | | Handle side | N.D. | N.D. | N.D. | N.D. | N.D. | N.D. | 2.8 | 2.7 | 4.4 | 2.3 | N.D. | N.D. | 33.3 |
| | Dehider | Edge side | N.D. | N.D. | N.D. | N.D. | N.D. | N.D. | N.D. | N.D. | N.D. | N.D. | N.D. | N.D. | 0.0 |
| | | Handle side | N.D. | N.D. | 5.1 | N.D. | N.D. | N.D. | N.D. | N.D. | N.D. | N.D. | N.D. | N.D. | 16.7 |
| Clean zone | Brisket saw | Edge side | N.D. | N.D. | N.D. | N.D. | N.D. | N.D. | N.D. | N.D. | N.D. | N.D. | N.D. | N.D. | 0.0 |
| | | Handle side | 6.1 | N.D. | 5.3 | N.D. | 7.7 | N.D. | N.D. | N.D. | 3.1 | N.D. | N.D. | N.D. | 66.7 |
| | Splitting saw | Edge side | N.D. | 2.6 | 1.8 | 2.0 | N.D. | N.D. | N.D. | N.D. | N.D. | N.D. | N.D. | N.D. | 33.3 |
| | | Handle side | N.D. | N.D. | 2.6 | N.D. | N.D. | N.D. | N.D. | N.D. | N.D. | N.D. | N.D. | N.D. | 16.7 |
| | Milling cutter | Edge side | N.D. | N.D. | N.D. | 2.2 | N.D. | N.D. | 7.4 | N.D. | N.D. | N.D. | 6.4 | N.D. | 50.0 |
| | | Handle side | N.D. | N.D. | N.D. | N.D. | N.D. | N.D. | N.D. | N.D. | N.D. | N.D. | N.D. | N.D. | 0.0 |
| | Knife | Edge side | N.D. | N.D. | N.D. | N.D. | N.D. | N.D. | N.D. | N.D. | N.D. | N.D. | 5.6 | N.D. | 16.7 |
| | | Handle side | N.D. | N.D. | N.D. | N.D. | 7.4 | N.D. | N.D. | N.D. | N.D. | N.D. | 6.8 | N.D. | 33.3 |
| Both zone | Apron* | - | N.D. | N.D. | 5.5 | 3.7 | 6.3 | N.D. | 1.4 | N.D. | N.D. | N.D. | N.D. | N.D. | 50.0 |
| | Trolley hook | - | 2.4 | N.D. | 4.2 | N.D. | 2.0 | 1.8 | N.D. | N.D. | N.D. | N.D. | 2.5 | N.D. | 66.7 |

*Bacteria counts were calculated as log CFU/cm$^2$

** *E. coli* and coliform bacteria were not detected in any of the trials from the equipment used in the slaughtering lines.

used in the slaughtering line, the brisket saw (handle side) and trolley hook were confirmed to be contaminated with microorganisms in the majority of trials, and the positive rate of microorganisms was 66.7% (4 times was positive among 6 time sampling). The microorganism adhesion rate of milling cutters (edge side) and aprons was 50.0%, while those of the foot cutter (edge and handle side), splitting saw (edge side), and knife (handle side) were 33.3%, and those of dehider (handle side), splitting saw (handle side), and knife (edge side) were 16.7%.

The Federation of Hida Meat Agricultural Cooperative Association, investigated in this study, was certified for FSSC22000 (certificate of registration no: JMAQA-FC126) and ISO22000: 2018 (certificate of registration no: JMAQA-F005) in the process of slaughtering and dismantling cattle and cutting carcasses. At this facility, wiping inspections of the processing line and equipment are performed once a week after cleaning as part of hygiene management to evaluate cleaning efficiency. The inspection targets medium-temperature bacteria, which can stimulate growth at 30–37°C. If contamination with these bacteria is confirmed, the cleaning operator is alerted, and thorough cleaning and microbial inspection are carried out until no microbial adhesion is detected. By monitoring the results of the wiping inspection for each cleaning operation, it is possible to confirm whether the hygiene management is adequate. Since the medium-temperature bacteria targeted here include food-poisoning bacteria such as pathogenic *E. coli* and *Salmonella*, they are critical for food safety. Coliforms were not detected in any of the sampling trials conducted in this study's equipment or lines. However, residual microorganisms were found in many samples (Table 1). As a result, it was proposed that hygiene management monitoring of psychrophilic bacteria and medium temperature bacteria can be fully utilized to evaluate the cleaning effect. Furthermore, during long-term refrigeration, psychrophilic bacteria can grow in beef and cause spoilage. If psychrophilic bacteria can be removed entirely from the equipment and lines through daily cleaning, it is considered that the adhesion of microorganisms to the carcass can be reduced, allowing the beef's expiration date to be extended.

The foot cutter and dehider are used in the dirty zone of the slaughtering line. At the beginning of the slaughter process, a foot cutter was used to cut the cattle's paw, and microbial loads of 1.4 to 2.9 log CFU / mL (Trial-2, 5) on TSA medium and 1.7 log CFU / mL (Trial 5) on MRS medium were confirmed on the edge side (Table 1). Furthermore, a microbial load of 2.8 to 4.4 log CFU / mL (Trial-4, 5) was confirmed in TSA medium and 2.3 to 2.7 log CFU / mL (Trial-4, 5) in MRS medium on the handle side of the foot cutter. After the cattle carcass is suspended, a dehider is used to skin the body. Bacteria were not detected from the edge side of the dehider, and 5.1 log CFU / mL was detected in TSA medium only in Trial 2 from the handle side. The microbial load attached to the hide and paw of cattle is high [3, 13]. Therefore, it is assumed that the number of bacteria adhering to the work equipment in the dirty zone was also high. The dehider has a lower microorganism detection rate than the foot cutter, and it is thought that the dehider has a lower chance of directly touching the hide during the skinning process. In addition, in the facility used in this experiment, the edges of knives and saws were sterilized by soaking in hot water at 83°C after each treatment. Although hot water disinfection is also performed on the foot cutter, it is hypothesized that the dehider has less microbial contamination accumulating during the work. As a result, the microbial contamination may be lower after cleaning. In the first step of the slaughtering line, there are many opportunities to come into contact with highly contaminated cattle parts such as paws and hides, and the number of bacteria after washing was high. This type of equipment requires thorough cleaning to prevent microbial contamination.

Next, we concentrated on the equipment used in the slaughtering line's clean zone (Brisket saw, splitting saw, milling cutter, and knife). Brisket saws are used to make cuts near the sternum of a cow, and no bacteria were detected from the edge side in any of the trials. On the

handle side, high microbial contamination was confirmed in the TSA medium, with the number of bacteria ranging from 3.1 to 7.7 log CFU / mL (Trial-1, 2, 3, 5). The splitting saw is a machine that divides whole beef carcasses in half, and the microbial counts of the edge side are 1.8 log CFU / mL (Trial-2) in TSA medium and 2.0 ~ 2.6 log CFU / mL (Trial-1, 2) in MRS medium. On the handle side, a bacterial count of 2.6 log CFU / mL (Trial-1) was recorded on TSA medium. A milling cutter is a machine used to remove the dura mater of the carcass after it has been split. On the edge side, a bacterial count of 2.2 log CFU / mL (Trial-2) was confirmed in MRS medium, and a high bacterial count of 6.4 to 7.4 log CFU / mL (Trial-4, 6) was detected in TSA medium. On the other hand, the number of bacteria on the handle side was below the detection limit in all trials. Knives are typically used for carcass trimming, with 5.6 log CFU / mL detected (Trial-6) in TSA medium on the edge side and 6.8 ~ 7.4 log CFU / mL (Trial-3, 6) on TSA medium on the handle side. A large number of microorganisms were detected on both sides of the knife.

The parts of the instrument where microbial contamination is likely to remain vary. While brisket saws and knives tend to have contamination on the handle side, splitting saws and milling cutters tend to have contamination on the edge side. The shape of the device may be related to the ease of cleaning. The brisket saw's cutting edge is relatively simple, whereas the shape of the cutting edge of the splitting saw and milling cutter is complicated, and meat residue is easily caught (S1 Fig). This suggests that it is necessary to select an appropriate cleaning method for its shape when cleaning equipment. In addition, for equipment such as brisket saws and knives that tend to remain contaminated on the handle side, measures such as disinfecting after cleaning on the handle side are required.

Bacterial counts of 1.4 to 6.3 log CFU / $cm^2$ (Trial-2, 3, 4) on TSA medium and 3.7 log CFU / $cm^2$ (Trial 2) on MRS medium were measured from the aprons worn by workers on the slaughter line. In particular, Trial 3 had a high bacterial count. Each time, a swab was taken from the aprons of randomly selected workers, but the detection rate was as high as 50%, and the number of bacteria also large. Aprons can be easily washed in a washing machine compared to other equipment. Therefore, it was determined that the bacterial adhesion to the apron was caused by improper cleaning, and thorough employee guidance was required. The trolley hook is used to hook the stunned cattle's leg and suspend the carcass, and its bacterial count ranges from 2.0 to 4.2 log CFU / mL (Trial-1, 2, 3, 6) in TSA medium and 1.8 log CFU / mL in MRS medium. The bacterial detection rate was the highest at 66.7%, while the number of bacteria was relatively low.

**Boning line.**    Compared to the slaughtering line, the distribution of microorganisms on the boning line was mainly concentrated on the electric saw (handle side) (Table 2). In Trial 6, only 0.8 log CFU / $cm^2$ were detected in the TSA medium from the turntable. An electric saw is a machine used to cut up carcasses into smaller parts, and no bacteria were detected from the edge side in any of the trials. On the other hand, from the handle side, 2.6 to 5.3 log CFU / mL (Trial-1, 4, 7) was detected in TSA medium, and 1.7 to 5.3 log CFU / mL was detected in MRS medium (Trial-1, 4, 5). The electric saw can be cleaned by removing the blade, but the main body, including the handle, cannot be washed with water (S2 Fig). Therefore, it was difficult to remove the microbial contamination accumulated during the work, and it was estimated that the detection rate of microorganisms was as high as 57.1%.

## Identification of psychrophilic microorganisms isolated from equipment and lines

**Slaughtering line.**    *Candida zeylanoides* was identified in all isolated colonies on the edge side of the foot cutter used in the dirty zone of the slaughtering line (Table 3). In Trial 5, all

**Table 2. The number of psychrophilic microorganisms adhering to equipment used in boning lines.**

| Sampling point | Bacterial counts in each medium (log CFU/ml) ** | | | | | | | | | | | | | | Positive rate (%) |
|---|---|---|---|---|---|---|---|---|---|---|---|---|---|---|---|
| | Trial1 | | Trial2 | | Trial3 | | Trial4 | | Trial5 | | Trial6 | | Trial7 | | |
| | 2019/8/7 | | 2019/12/12 | | 2019/12/18 | | 2020/7/29 | | 2020/8/5 | | 2021/1/7 | | 2021/1/13 | | |
| | TSA | MRS | TSA | MRS | TSA | MRS | TSA | MRS | TSA | MRS | TSA | MRS | TSA | MRS | |
| Large division conveyor belt* | N.D. | N.D. | N.D. | N.D. | N.D. | N.D. | N.D. | N.D. | N.D. | N.D. | N.D. | N.D. | N.D. | N.D. | 0.0 |
| Boning conveyor belt* | N.D. | N.D. | N.D. | N.D. | N.D. | N.D. | N.D. | N.D. | N.D. | N.D. | N.D. | N.D. | N.D. | N.D. | 0.0 |
| Turntable* | N.D. | N.D. | N.D. | N.D. | N.D. | N.D. | N.D. | N.D. | N.D. | N.D. | 0.8 | N.D. | N.D. | N.D. | 14.3 |
| Meat holder | N.D. | N.D. | N.D. | N.D. | N.D. | N.D. | N.D. | N.D. | N.D. | N.D. | N.D. | N.D. | N.D. | N.D. | 0.0 |
| Electric saw-Edge side | N.D. | N.D. | N.D. | N.D. | N.D. | N.D. | N.D. | N.D. | N.D. | N.D. | N.D. | N.D. | N.D. | N.D. | 0.0 |
| Electric saw-handle side | 5.3 | 5.3 | N.D. | N.D. | N.D. | N.D. | 3.0 | 2.8 | N.D. | 1.7 | N.D. | N.D. | 2.6 | N.D. | 57.1 |

* Bacteria counts were calculated as log CFU/cm$^2$

** *E. coli* and coliform bacteria were not detected in any of the trials from the equipment used in the boning lines.

isolated colonies on the handle side were identified as *C. zeylanoides*. *Pseudomonas* sp., *Chryseobacterium* sp., *Meyerozyma guilliermondii*, and *Rhodotorula* sp., on the other hand, were identified as microorganisms that remained on the handle side of the foot cutter in Trial 4. According to these findings, *C. zeylanoides* were mainly distributed in the foot cutter regardless of the edge or handle sides. Yeasts such as *M. guilliermondii* and *Rhodotorula* sp. were also found. Previous studies have also found yeasts such as *C. zeylanoides* and *Rhodotorula* sp. isolated from slaughterhouse lines and equipment [14]. These yeasts are known to be derived from soil and livestock [15]. In addition, all isolated colonies from the dehider used in the cattle skinning process were identified as *Rhodococcus* sp. is a gram-positive bacterium that is widely distributed on grazing farms and is frequently isolated from the feces of wild animals and livestock, such as pigs, cattle, and horses [16]. Thus, microorganisms derived from cattle and soil are mainly detected in the equipment used in the dirty zone of the slaughtering line. It has been demonstrated that they are not entirely removed by the washing process and remain on the equipment's surface.

Isolated colonies on the handle side of the brisket saw used in the clean zone were identified as *Pseudomonas* sp. in all samples. Yeasts such as *C. zeylanoides*, *Rhodotorula* sp., and *Candida parapsilosis* were frequently found on the edge side of the splitting saw (for splitting carcasses), as well as on the foot cutter. Therefore, it was considered that yeast may have survived on the edge side of the splitting saw after washing. In addition, three of the four colonies isolated on the handle side of the splitting saw were *C. zeylanoides*, indicating that yeast contamination was distributed on both sides. On the edge side of the milling cutter, *C. parapsilosis* was predominant in Trial 2, while *Pseudomonas* sp. was predominant in Trials 4 and 6. All colonies isolated from the edge and handle sides of the knife used for carcass trimming were identified as *Pseudomonas* sp.

Although this sampling was performed after the washing treatment, yeasts such as *Rhodotorula* sp. and *C. zeylanoides* and bacteria including *Moraxella osloensis* and *Pseudomonas* sp. were detected from the edge side of the splitting saw, milling cutter, and knife. So far, cleaning plans have focused on gram-negative bacteria that cause food poisoning, but it has been demonstrated that environmental microorganisms, such as yeast, may not be entirely removed by the cleaning process and may remain on the equipment.

Since the edge side of the saw comes into direct contact with the carcass, there is a high possibility that microbial contamination will spread from the carcass. Therefore, it was

**Table 3. Identification results of psychrophilic microorganisms adhering to equipment used in slaughtering line.**

| Zone where equipment is used | Sampling point | Sampling side | Trial | Isolation medium | Identified genus/ species | The number of isolated colonies |
|---|---|---|---|---|---|---|
| Dirty zone | Foot cutter | Edge side | Trial 2 | TSA | *Candida zeylanoides* | 4/4 |
| | | | Trial 5 | TSA | *Candida zeylanoides* | 4/4 |
| | | | | MRS | *Candida zeylanoides* | 4/4 |
| | | Handle side | Trial 4 | TSA | *Pseudomonas* sp. | 1/4 |
| | | | | | *Chryseobacterium* sp. | 1/4 |
| | | | | | *Meyerozyma guilliermondii* | 1/4 |
| | | | | | *Rhodotula* sp. | 1/4 |
| | | | | MRS | *Rhodotula* sp. | 4/4 |
| | | | Trial 5 | TSA | *Candida zeylanoides* | 4/4 |
| | | | | MRS | *Candida zeylanoides* | 4/4 |
| | Dehider | Handle side | Trial 2 | TSA | *Rhodococcus* sp. | 4/4 |
| Clean zone | Brisket saw | Handle side | Trial 1 | TSA | *Pseudomonas* sp. | 4/4 |
| | | | Trial 2 | TSA | *Pseudomonas* sp. | 4/4 |
| | | | Trial 3 | TSA | *Pseudomonas* sp. | 4/4 |
| | | | Trial 5 | TSA | *Pseudomonas* sp. | 4/4 |
| | Splitting saw | Edge side | Trial 1 | MRS | *Rhodotorula* sp. | 4/4 |
| | | | Trial 2 | TSA | *Candida zeylanoides* | 3/4 |
| | | | | | *Moraxella osloensis* | 1/4 |
| | | | | MRS | *Candida zeylanoides* | 4/4 |
| | | Handle side | Trial 2 | TSA | *Candida zeylanoides* | 3/4 |
| | | | | | *Rhodococcus* sp. | 1/4 |
| | Milling cutter | Edge side | Trial 2 | MRS | *Candida parapsilosis* | 4/4 |
| | | | Trial 4 | TSA | *Pseudomonas* sp. | 4/4 |
| | | | Trial 6 | TSA | *Pseudomonas* sp. | 4/4 |
| | Knife | Edge side | Trial 6 | TSA | *Pseudomonas* sp. | 4/4 |
| | | Handle side | Trial 3 | TSA | *Pseudomonas* sp. | 4/4 |
| | | | Trial 6 | TSA | *Pseudomonas* sp. | 4/4 |
| Both zone | Apron | - | Trial 2 | TSA | *Pseudomonas* sp. | 4/4 |
| | | | | MRS | *Yarrowia galli* | 4/4 |
| | | | Trial 3 | TSA | *Pseudomonas* sp. | 4/4 |
| | | | Trial 4 | TSA | *Acinetobacter junii* | 1/4 |
| | | | | | *Pseudomonas* sp. | 3/4 |
| | Trolley hook | - | Trial 1 | TSA | *Rhodococcus* sp. | 4/4 |
| | | | Trial 2 | TSA | *Rhodococcus* sp. | 4/4 |
| | | | Trial 3 | TSA | *Candida zeylanoides* | 4/4 |
| | | | | MRS | *Sporidiobolus salmomnicolor* | 2/4 |
| | | | | | *Candida zeylanoides* | 2/4 |
| | | | Trial 6 | TSA | *Candida zeylanoides* | 4/4 |

determined that the cleaning and disinfection method for the edge side needed to be changed to completely remove microorganisms, including yeast. *Pseudomonas* sp. was found in all samples from the aprons used in the slaughtering process and bacterial groups such as *Yarrowia galli* and *Acinetobacter jejunii*. Trolley hooks have been found to contain a wide variety of microorganisms, including *C. zeylanoides*, *Rhodococcus* sp., and *Sporidiobolus salmonicolor*. With the exception of the brisket saw-handle side of Trial-5, all four colonies isolated from each sample were identified as *Pseudomonas* sp. (Brisket saw-handle side, milling cutter-edge side, knife-both side, apron), and the bacterial count was as high as 5.3 log CFU / mL or more.

If *Pseudomonas* with a high number of bacteria is present on the surface of the equipment, a biofilm may form. Previous research has shown that when *Pseudomonas* forms a biofilm on the surface of a piece of equipment, it promotes pathogen colonization and activity [17, 18]. Furthermore, other studies have found that *Pseudomonas* sp. was frequently isolated from the instruments used in cattle and sheep slaughter lines, even after they had been washed [19]. Previously, we conducted a long-term preservation test of beef carcasses slaughtered and dismantled at this slaughterhouse [8]. Under aerobic storage conditions, the bacterial counts were reached at 8–9 log CFU/g, and *Pseudomonas* spp. predominated the beef microbiota at nine weeks. Numerous studies reported that *Pseudomonas* spp. is related to meat spoilage under low-temperature conditions [20–23]. Hence, we consider it necessary to remove *Pseudomonas* spp. from the equipment by washing because of its putrefactive activity.

This study did not investigate whether microorganisms present in the meat are the same as those identified on the equipment and surfaces. To clarify this, strain identification using molecular typing methods, such as RAPD, MLST, and ribotyping, need to be performed [24–29]. Further detailed research should be carried out in the future to determine what kinds of equipment and bacterial species are likely to contaminate the surface of carcasses.

**Boning line.** All isolated colonies from the turntable on the boning line were identified as *C. zeylanoides* (Table 4). In addition, *C. zeylanoides* were identified on the electric saw's handle side in all trials except Trial 7. In contrast, all *Pseudomonas sp.* colonies were identified in Trial 7. Only *C. zeylanoides* and *Pseudomonas* sp. were detected in the boning line, whereas numerous yeast species were found in the slaughtering line. It is presumed that the reason for this is that a large number of contaminated microorganisms derived from living organisms such as the hide and feces are introduced into the slaughtering line, and the number and types of bacteria adhering to the equipment are numerous. The boning line, on the other hand, is the process of removing bone and shaping the carcass. It is believed that the majority of the bacteria brought in are derived from bacteria attached to the carcass after slaughter.

In the past, the distribution of spoilage yeasts was investigated with instruments in wineries, bakery industries, breweries, yogurt factories or goat cheese industries [30–34]. The processed meats industry also investigated the distribution of yeast across the instruments used in the industry and identified the surface of facilities, including room equipment and production materials, as the main source of yeast contamination of cured meat [35]. In particular, *Candida zeylanoides* is frequently isolated from cured meat, and considered predominant in the raw material, fresh meat. In this study, *Candida zeylanoides* was frequently isolated from the equipment used in the boning line, suggesting that the slaughterhouse, which is the most upstream food distribution center, may be a source of yeast contamination of meat.

Microbial contamination was concentrated on the handle side of the electric saw, where microbes were frequently detected in multiple trials. The concentration of disinfectants,

**Table 4. Identification results of psychrophilic microorganisms adhering to equipment used in the boning line.**

| Sampling point | Trial | Isolation medium | Identified genus/ species | The number of isolated colonies |
|---|---|---|---|---|
| Turntable | Trial 6 | TSA | *Candida zeylanoides* | 4/4 |
| Electric saw-handle side | Trial 2 | TSA | *Candida zeylanoides* | 4/4 |
| | | MRS | *Candida zeylanoides* | 4/4 |
| | Trial 4 | TSA | *Candida zeylanoides* | 4/4 |
| | | MRS | *Candida zeylanoides* | 4/4 |
| | Trial 5 | MRS | *Candida zeylanoides* | 4/4 |
| | Trial 7 | TSA | *Pseudomonas* sp. | 4/4 |

required to remove yeast, was higher than that of food-related bacteria [36]. Salo et al. (2005) demonstrated that alcohol-based disinfectants were most effective in decontaminating yeast isolates [37]. In addition, surfactant-based and peroxide-based disinfectants were effective against floating yeast cells, while biofilm carrier tests reported the effectiveness of chlorine-based foam cleaners. On the other hand, disinfectants containing chlorine and persulfates have failed to kill yeast cells in both suspensions and biofilm formation. As the susceptibility of bacteria and yeast to disinfectants is different, a thorough review of cleaning methods for yeast contaminated equipment is warranted.

## Conclusion

The purpose of this study was to determine the distribution of psychrophilic microorganisms that remained on the equipment and lines in the slaughtering and boning lines after cleaning. Many of the equipment used in the slaughtering line have complicated shapes. The microbial contamination varies depending on the site, the attached microorganisms are diverse, and they are environment-derived microorganisms. On the other hand, on the boning line, the equipment with residual microbial contamination and their bacterial species showed the same tendency in multiple trials. It is thought that providing information to the meat industry on equipment and parts where microorganisms are difficult to remove by washing treatment can further improve hygiene management.

## Supporting information

**S1 Fig. Photograph of the equipment used in the slaughtering line.**
(TIF)

**S2 Fig. Photograph of the equipment used in boning line.**
(TIF)

## Acknowledgments

We would like to thank Editage (www.editage.com) for English language editing.

## Author Contributions

**Conceptualization:** Ayaka Nakamura, Hajime Takahashi.

**Investigation:** Ayaka Nakamura, Anrin Kondo, Fumiaki Koike.

**Methodology:** Ayaka Nakamura.

**Resources:** Takashi Kuda, Bon Kimura, Mitsushi Kobayashi.

**Supervision:** Hajime Takahashi, Takashi Kuda, Bon Kimura, Mitsushi Kobayashi.

**Writing – original draft:** Ayaka Nakamura, Anrin Kondo.

**Writing – review & editing:** Hajime Takahashi.

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
