## [Decision Letter · Decision Letter 0]

23 Feb 2022

PONE-D-22-01666Distribution of psychrophilic microorganisms in the beef slaughterhouse after cleaningPLOS ONE

Dear Dr. Takahashi,

Thank you for submitting your manuscript to PLOS ONE. After careful consideration, we feel that it has merit but does not fully meet PLOS ONE’s publication criteria as it currently stands. Therefore, we invite you to submit a revised version of the manuscript that addresses the points raised during the review process.

Pay special attention to the suggestions done regarding the objective and hypothesis presented in the document.

We look forward to receiving your revised manuscript.

Kind regards,

Guadalupe Virginia Nevárez-Moorillón, Ph.D.

Academic Editor

PLOS ONE

Journal Requirements:

Reviewers' comments:

Reviewer's Responses to Questions

5. Review Comments to the Author

Reviewer #1: The manuscript “Distribution of psychrophilic microorganisms in the beef slaughterhouse after cleaning” aimed to explore the microbial contamination present in a beef slaughterhouse in Japan. The topic is definitely relevant and warranted, but the manuscript is very poorly written. It needs substantial improvements, both on the scientific language and structure.

The goal of the study cannot be met with this experimental design, and the study also lacks discussion of the results – only 18 references given.

Abstract

L19: give details on the methods used to determine microbial contamination

L22: “Petri dish”… how did you select the colonies?

L24: WGS or other sequencing technology? If you also looked for yeasts, you need to correct “the bacterial species” that you mention before.

L27: leave this conclusion for the end… where do you think the overall high contamination comes from and how did you investigate environmental contamination?

Introduction

L41: what about Japan? What is the status quo? You keep mentioning the international point-of-view, but since your samples were collected in Japan (an in only one facility), you should start from there and then compare with the international practices.

L62: cannot be stopped? Then why do we do it?

L66: remove “to zero”

Materials and Methods

L86: “Materials”

L89: why not before?

L93: what is the wiping inspection?

L87: Please provide a schematic figure with all sampling points.

L108: what is the difference between these media e.g., what are they selective for?

L110: which dilution did you plate?

L136: “)” missing

L120: please split this in two subheadings

Results and Discussion

L164: how did you calculate the adhesion rate?

How do you explain the distribution of Pseudomonas sp.? Additionally, do you mean sp. or spp.?

Reviewer #2: The objective of this study was to investigate the abundance and distribution of psychrophilic microorganisms associated with spoilage in beef slaughterhouse environments after cleaning. The topic is interesting, but there are two relevant aspects that weaken the work. The count of psychrophilic was carried out in different sectors of a slaughterhouse, but this does not imply that they are associated with meat contamination and its subsequent alteration. Since the psychrophilic microorganisms present in the meat were not identified, it was not possible to associate whether the contamination found in the slaughterhouse environment had any relevance to the meat quality. On the other hand, the number of colonies isolated for later identification was low and it is not possible to establish the diversity of microorganisms present or their load on the equipment.

Line 30: It is concluded that, despite the cleaning system used in the slaughterhouse, the concentration of psychrophilic was "very high". However, no parameters are given to support that this load was "very high".

The introduction seems to be very long. Authors should focus on the most important aspects that support their work.

Lines 81-82: "The ultimate goal of this study was to reduce contamination of carcasses by microorganisms from slaughter processing lines and equipment". This objective was not addressed in this work.

Lines 190-192: "If psychrophilic bacteria can be removed entirely from the equipment and lines through daily cleaning, it is considered that the adhesion of microorganisms to the carcass can be reduced, allowing the beef’s expiration date to be extended". It is quite obvious that if microorganisms are completely removed from equipment and surfaces in slaughterhouses, contamination of meat will be greatly reduced. However, the goal of a cleaning system is not sterilization but the reduction to acceptable levels. Therefore, observed microorganisms concentration should be compared with expected concentrations to determine their impact. This is not addressed in this work.

Lines 275-276: It is M&M.

Line 321: It just remains to check if the microorganisms present in the meat are the same as those identified in the equipment and surfaces. So, It seems to be a hypothesis (logical) or a speculation.

---

## [Author Response · Author response to Decision Letter 0]

17 Mar 2022

Response to Reviewers

Thank you for inviting us to submit a revised draft of our manuscript entitled, “Distribution of psychrophilic microorganisms in the beef slaughterhouse after cleaning” to PLOS ONE. We also appreciate the time and effort you and each of the reviewers have dedicated to providing insightful feedback on ways to strengthen our paper. Thus, it is with great pleasure that we resubmit our article for further consideration. We have incorporated changes that reflect the detailed suggestions you have graciously provided. We also hope that our edits and the responses we provide below satisfactorily address all the issues and concerns the reviewers have noted.

Dear Reviewer #1:

Abstract

L19: give details on the methods used to determine microbial contamination

(Response)

Thank you for the suggestion.

As recommended, I added detailed information about the microbial investigation.

(Page 2, Line 18–21): The processing lines and equipment used in slaughtering and boning were swabbed, and the microbial count was determined by the culture method using a TSA and MRS medium and Chromocult® Coliform agar incubated at 15ºC and 37ºC, respectively. 

L22: “Petri dish”… how did you select the colonies?

(Response)

Thank you for your question. I investigated the predominant microbial species attached to the equipment’s surface. I randomly selected four colonies from one plate. I added detailed information to the original manuscript as follows:

(Page 2, Line 23–25) Next, four colonies were randomly isolated from the one petri dish used for the bacterial count measurement to identify the predominant microbial species of the microorganisms attached to each equipment.

L24: WGS or other sequencing technology? If you also looked for yeasts, you need to correct “the bacterial species” that you mention before.

(Response)

Thank you for your accurate instruction. I used sanger sequencing technology and I added the information to the manuscript (Page 2, Line 26). Also, I changed “the bacterial species” to “the microbial species” (Page 2, Line 25). 

L27: leave this conclusion for the end… where do you think the overall high contamination comes from and how did you investigate environmental contamination?

(Response)

Thank you for your question. This study did not investigate environmental contamination. However, the Discussion section cites that these yeasts and bacteria were isolated from soil and livestock (Page 20, Line 286–92). 

Introduction

L41: what about Japan? What is the status quo? You keep mentioning the international point-of-view-, but since your samples were collected in Japan (an in only one facility), you should start from there and then compare with the international practices.

(Response)

Thank you for your insightful feedback. In fact, in Japan, introducing HACCP became mandatory in slaughterhouses from June 2021. I should have included information about this in the manuscript. Thank you for pointing it out. I modified the document as follows:

(Page 4, Line 42–45) Process control by the Hazard Analysis and Critical Control Point (HACCP) system is one measure to prevent the contamination of carcasses by such harmful bacteria. In Japan, following the EU and the United States, introducing the HACCP system in slaughterhouses became mandatory from June 2021. 

L62: cannot be stopped? Then why do we do it?

(Response)

I'm sorry that my previous sentence lacked an explanation. Beef chilled shipping exports can be transported while preserving beef quality compared to frozen conditions. However, low-temperature-growing microorganisms can grow and cause quality degradation in beef. It is important to reduce microorganisms that adhere to beef as much as possible to prevent such a scenario. I corrected the sentence as follows:

(Page 5, Page 63–65) However, long transportation periods can allow some low-temperature-growing microorganisms to grow and degrade beef quality. 

L66: remove “to zero”

(Response)

Thank you for your instruction. I remove “to zero”.

Materials and Methods

L86: “Materials”

(Response)

I apologize for simple mistake. I corrected it.

L89: why not before?

(Response)

Thank you for your critical question.

This study aimed to investigate the distribution of psychrophilic microorganisms in the equipment used in the slaughtering and boning lines and identify the equipment on which microbial contamination is likely to remain after washing. Therefore, we targeted the equipment after cleaning. By feeding back information about difficult-to-clean equipment to the field workers, they can select an appropriate cleaning method and expect further improvement in hygiene management.

L93: what is the wiping inspection?

(Response)

I apologize. My previous sentence was not appropriate. I changed the sentence as follows:

(Page 7, Line 89-90) Cleaning was carried out after each line was used, and the equipment was wiped the day after cleaning.

L87: Please provide a schematic figure with all sampling points.

(Response)

Thank you for your suggestion. This study aimed to investigate the distribution of psychrophilic microorganisms that adhere to equipment after cleaning, and we do not discuss the transmission of microorganisms in slaughterhouse. Therefore, I decided that the schematic figure should not be included. Instead, we added a photo of the equipment targeted in this study to the supplementary figure. In the future, when conducting research such as strain typing to explore the dynamics of microorganisms in the slaughterhouse, I would like to attach a schematic figure as you suggested. Thank you for your valuable opinion.

L108: what is the difference between these media e.g., what are they selective for?

(Response)

Thank you for your question. I added the information regarding the media as follows: 

(Page 8, Line 110–114) The number of bacteria in each swab sample was measured on three types of media: trypticase soy agar (TSA; Becton, Dickinson and Company, Franklin Lakes, NJ, USA) for general bacteria, de Man, Rogosa, and Sharpe (MRS) agar (Merck KGaA, Darmstadt, Germany) for lactic acid bacteria, and Chromocult®︎ Coliform Agar (Merck) for coliform bacteria and E. coli.

L110: which dilution did you plate?

thank you for your question. I added the information as follows:

(Response)

(Page 8, Line 108-109) The swab sample was diluted ten-fold with saline solution, and 100 µL of each original solution and diluent were spread onto each agar medium.

L136: “)” missing

(Response)

Thank you for your polite point. I corrected it. 

L120: please split this in two subheadings

(Response)

Thank you for your suggestion. I split this paragraph in two subheadings.

Results and Discussion

L164: how did you calculate the adhesion rate?

(Response)

Thank you for your question.

We sampled the slaughtering and boning lines six and seven times, respectively. Adhesion rate (positive rate in Tables 1 and 2) indicates the ratio of the number of times that the number of bacteria above the detection limit was confirmed. Due to the previous sentence confusion, I modified it as follows: 

(Page 11, Line 161-164) Among the equipment used in the slaughtering line, the brisket saw (handle side) and trolley hook were contaminated with microorganisms in the majority of trials, and the positive rate of microorganisms was 66.7% (four trials were positive among six samplings).

How do you explain the distribution of Pseudomonas sp.? 

(Response)

Since Pseudomonas was widely distributed in this slaughterhouse, we presumed that the original contamination from the living body was large. In other studies I am conducting, Pseudomonas accounts for more than 80% of the flora on the carcass surface during the slaughter process. Hence, we assumed that Pseudomonas from the carcass contaminates the equipment and remains there after cleaning. 

Additionally, do you mean sp. or spp.?

(Response)

"sp." is used as the singular and "spp." is used as the plural. In this manuscript, all are written as "sp.".

 

Dear Reviewer #2: 

The objective of this study was to investigate the abundance and distribution of psychrophilic microorganisms associated with spoilage in beef slaughterhouse environments after cleaning. The topic is interesting, but there are two relevant aspects that weaken the work. The count of psychrophilic was carried out in different sectors of a slaughterhouse, but this does not imply that they are associated with meat contamination and its subsequent alteration. Since the psychrophilic microorganisms present in the meat were not identified, it was not possible to associate whether the contamination found in the slaughterhouse environment had any relevance to the meat quality. On the other hand, the number of colonies isolated for later identification was low and it is not possible to establish the diversity of microorganisms present or their load on the equipment.

(Response)

Thank you for your insightful comments.

In past studies, we conducted a long-term preservation test of beef carcasses slaughtered and dismantled at this slaughterhouse (Dynamics of microbiota in Japanese Black beef stored for a long time under chilled conditions. Food microbiology, 2021, 100: 103849). That study reported that Pseudomonas sp. was the predominant bacterial group in the meat when stored in aerobic packaging. It is well reported that the Pseudomonas genus is a psychrophilic spoilage-causing bacterium. This study revealed that Pseudomonas sp. was widely distributed in the equipment used in this slaughterhouse. From these results, we believe that the psychrophilic microorganisms identified in this study may adhere to the meat and are associated with spoilage. I added the following to the Discussion section: 

(Page 25, Line 333–339) Previously, we conducted a long-term preservation test of beef carcasses slaughtered and dismantled at this slaughterhouse (8). Under aerobic storage conditions, the bacterial counts were reached at 8–9 log CFU/g, and Pseudomonas sp. predominated the beef microbiota at nine weeks. Numerous studies reported that Pseudomonas sp. is related to meat spoilage under low-temperature conditions (20–23). Hence, we consider it necessary to remove Pseudomonas sp. from the equipment by washing because of its putrefactive activity.

I selected four colonies for later identification. Certainly, the resolution is low for investigating bacterial diversity, but it is sufficient for investigating the predominant bacterial group. There were many test plots in which this study identified all four strains of the four selected colonies as belonging to the same genus.

Line 30: It is concluded that, despite the cleaning system used in the slaughterhouse, the concentration of psychrophilic was "very high". However, no parameters are given to support that this load was "very high".

(Response)

Thank you for your opinion. As you mentioned, I should not use “highly contaminated”. I modified the sentence as follows:

(Page 2, Line 31–32) Despite the cleaning operations, this study identified some equipment contaminated with microorganisms.

The introduction seems to be very long. Authors should focus on the most important aspects that support their work.

(Response)

Thank you for your valuable opinion. PLOS ONE has no limit on the number of characters in the manuscript, and few people are familiar with the hygiene management of beef slaughterhouses, so I wrote the introduction with this in mind. I believe that all the paragraphs are important in explaining the background of this study. Some sentences were changed with the reader’s understanding in mind (Page 4, Line 42–45; Page 5, Line 63–65; Page 6, Line 80-82). It would be helpful if you could proceed without cutting the content.

Lines 81-82: "The ultimate goal of this study was to reduce contamination of carcasses by microorganisms from slaughter processing lines and equipment". This objective was not addressed in this work.

(Response)

Thank you for your critical opinion. The previous sentence was a leap forward. I modified the sentence as follows:

(Page 6, Line 80-82) The purpose of this study was to understand the actual contamination of low-temperature microorganisms in the equipment used in the slaughtering and boning lines.

Lines 190-192: "If psychrophilic bacteria can be removed entirely from the equipment and lines through daily cleaning, it is considered that the adhesion of microorganisms to the carcass can be reduced, allowing the beef’s expiration date to be extended". It is quite obvious that if microorganisms are completely removed from equipment and surfaces in slaughterhouses, contamination of meat will be greatly reduced. However, the goal of a cleaning system is not sterilization but the reduction to acceptable levels. Therefore, observed microorganisms concentration should be compared with expected concentrations to determine their impact. This is not addressed in this work.

(Response)

The cleaning system needs to reduce the number of bacteria on the equipment’s surface to below the detection limit. In fact, at this facility, they perform a wiping inspection of the equipment after cleaning once a week, targeting mesophilic bacteria (35–37ºC). If the number of bacteria is measured, they sanitize the equipment again. This study investigated psychrophilic bacteria and the number of Escherichia coli and coliform bacteria, which are mesophilic bacteria. We measured the number of psychrophilic bacteria but not mesophilic bacteria. From this result, it was assumed that psychrophilic bacteria are more likely to remain on the instrument’s surface than mesophilic bacteria. However, no instrument had a 100% detection rate of psychrophilic bacteria on the instrument’s surface after cleaning, indicating that the number of psychrophilic bacteria could be reduced to below the detection limit in all equipment. Therefore, this study regarded the value as unacceptable when the number of bacteria in the equipment after cleaning was above the detection limit.

Lines 275-276: It is M&M.

(Response)

Thank you for your instruction. I deleted the sentence.

Line 321: It just remains to check if the microorganisms present in the meat are the same as those identified in the equipment and surfaces. So, It seems to be a hypothesis (logical) or a speculation.

(Response)

Thank you for your opinion. As you instructed, I modified the sentence as follows:

(Page 25, Line 316-319) This study did not investigate whether microorganisms present in the meat are the same as those identified on the equipment and surfaces. However, since the edge side of the saw comes into direct contact with the carcass, there is a high possibility that microbial contamination will spread from the carcass.

---

## [Decision Letter · Decision Letter 1]

6 Apr 2022

PONE-D-22-01666R1Distribution of psychrophilic microorganisms in the beef slaughterhouse after cleaningPLOS ONE

Dear Dr. Takahashi,

Thank you for submitting your manuscript to PLOS ONE. After careful consideration, we feel that it has merit but does not fully meet PLOS ONE’s publication criteria as it currently stands. Therefore, we invite you to submit a revised version of the manuscript that addresses the points raised during the review process.

The manuscript has been improved, but still needs to be revised in the discussion section, as suggested by the reviewers. Please attend these suggestions.

We look forward to receiving your revised manuscript.

Kind regards,

Guadalupe Virginia Nevárez-Moorillón, Ph.D.

Academic Editor

PLOS ONE

Reviewers' comments:

Reviewer's Responses to Questions

6. Review Comments to the Author

Reviewer #1: Title: replace “the” by “a” … in Japan after cleaning

Abstract:

L19: remove “by the culture method”

L21: remove “in most trials”, otherwise you need to give before how many trials were conducted

L22: remove “confirmed to be”

L24: remove “one”

L26: Sanger, not sanger

Can you give the % of contamination or the number of samples e.g., (10/24) where you found the microorganisms? Gives an idea of the prevalence of contamination, which is an easy an d interesting info to extract directly from the Abstract

Results and discussion:

This section is very poorly discussed. There is indeed a. lot of detail given, but barely any discussion with the available literature. This is not a report, it is a research article.

L335-339: Pseudomonas should be italicized. This section also reflects my previous comment: it should be spp., as you would expect several species of Pseudomonas to be found, not just one specific species (for those cases, use sp., when you are referring to a specific Pseudomonas).

Reviewer #2: The authors have adequately addressed my previous comments. Although I have a disagreement with the interpretation of some points, it is not a sufficient reason not to accept the modifications made. I only have one comment left that I consider to be a weakness of the study but to be resolved in future works.

I understand and agree with the authors that Pseudomonas are the most representative populations of psychrophilic microorganisms in meat. However, from this study, you cannot assume that the microorganisms present in the equipment are the same as those on the carcasses. For this, it should resort to molecular studies.

---

## [Author Response · Author response to Decision Letter 1]

28 Apr 2022

Response to Reviewers

Thank you for inviting us to submit a revised draft of our manuscript entitled, “Distribution of psychrophilic microorganisms in a beef slaughterhouse in Japan after cleaning” to PLOS ONE. We also appreciate the time and effort you and each of the reviewers have dedicated to providing insightful feedback on ways to strengthen our paper. Thus, it is with great pleasure that we resubmit our article for further consideration. We have incorporated changes that reflect the detailed suggestions you have graciously provided. We also hope that our edits and the responses we provide below satisfactorily address all the issues and concerns the reviewers have noted.

Dear Reviewer #1:

Thank you for your detailed suggestions regarding the Abstract. Thanks to these, the text has been vastly improved. With regard to the Discussion section, because there are currently no published reports of studies similar to the one we have carried out, it was difficult to compare and discuss our results with other findings side by side. However, we have deepened the discussion as much as possible. Thank you for pointing this out.

Title: replace “the” by “a” … in Japan after cleaning

(Response)

Thank you for the suggestion. As recommended, I modified the title.

Abstract:

L19: remove “by the culture method”

(Response)

Thank you for your instruction. I removed “by the culture method”

L21: remove “in most trials”, otherwise you need to give before how many trials were conducted 

(Response)

Thank you for your instruction. I removed “in most trials”

L22: remove “confirmed to be”

(Response)

Thank you for your instruction. I removed “confirmed to be”

L24: remove “one”

(Response)

Thank you for your instruction. I removed “one”

L26: Sanger, not sanger

(Response)

Thank you for your correction. I corrected it.

Can you give the % of contamination or the number of samples e.g., (10/24) where you found the microorganisms? Gives an idea of the prevalence of contamination, which is an easy and interesting info to extract directly from the Abstract

(Response)

Thank you for your suggestion. I have added the information about the percentages of contamination, as indicated below. Thanks to you, the Abstract text is now better.

Page 2, Lines 21–25: As a result, the brisket saw (handle side) and trolley hook were the most heavily contaminated with microorganisms, with each having a microbial adhesion rate of 66.7%. The microbial adhesion rates of the apron and milling cutter (edge side) were 50%, respectively, and those of the foot cutter (edge and handle side), splitting saw (edge side), and knife (handle side) were 33.3%, respectively.

Results and discussion:

This section is very poorly discussed. There is indeed a lot of detail given, but barely any discussion with the available literature. This is not a report, it is a research article.

(Response)

Thank you for your helpful suggestions. To the best of our knowledge, there are no published papers on studies that have investigated the distribution of psychrophilic microorganisms on different slaughterhouse equipment after cleaning. Therefore, a discussion based on available literature findings was difficult. However, we have deepened our discussion on the following aspects: why contamination was high in the specific equipment (Lines: 206–218, 238–246, 268-271), the type of bacteria attached (Lines: 286–295, 353–361), and which of those bacteria are involved with putrefactive risks (Lines: 330-342, 361-364). Additionally, in response to your suggestions, we have added following discussion section regarding yeast contamination of food industry and yeast susceptibility against disinfectant. Thank you for your valuable opinion.

Page 27, Line 364-373: In the past, the distribution of spoilage yeasts was investigated with instruments in wineries, bakery industries, breweries, yogurt factories or goat cheese industries. The processed meats industry also investigated the distribution of yeast across the instruments used in the industry and identified the surface of facilities, including room equipment and production materials, as the main source of yeast contamination of cured meat [35]. In particular, Candida zeylanoides is frequently isolated from cured meat, and considered predominant in the raw material, fresh meat. In this study, Candida zeylanoides was frequently isolated from the equipment used in the boning line, suggesting that the slaughterhouse, which is the most upstream food distribution center, may be a source of yeast contamination of meat.

Page 28, Line 375-385: Microbial contamination was concentrated on the handle side of the electric saw, where microbes were frequently detected in multiple trials. The concentration of disinfectants, required to remove yeast, was higher than that of food-related bacteria [36]. Salo et al. (2005) demonstrated that alcohol-based disinfectants were most effective in decontaminating yeast isolates [37]. In addition, surfactant-based and peroxide-based disinfectants were effective against floating yeast cells, while biofilm carrier tests reported the effectiveness of chlorine-based foam cleaners. On the other hand, disinfectants containing chlorine and persulfates have failed to kill yeast cells in both suspensions and biofilm formation. As the susceptibility of bacteria and yeast to disinfectants is different, a thorough review of cleaning methods for yeast contaminated equipment is warranted.

L335-339: Pseudomonas should be italicized. This section also reflects my previous comment: it should be spp., as you would expect several species of Pseudomonas to be found, not just one specific species (for those cases, use sp., when you are referring to a specific Pseudomonas).

(Response)

Thank you for pointing out these errors. I apologize for overlooking the italicization of bacterial names and this has since been corrected. Also, as you pointed out, when multiple Pseudomonas species are being described, they are represented as “spp.” 

Dear Reviewer #2:

I understand and agree with the authors that Pseudomonas are the most representative populations of psychrophilic microorganisms in meat. However, from this study, you cannot assume that the microorganisms present in the equipment are the same as those on the carcasses. For this, it should resort to molecular studies.

(Response)

We agree totally with what you say. It is necessary to use molecular typing methods (MLST, RAPD, ribotyping, etc.) to determine whether the bacteria attached to the equipment actually propagate to the carcass surface. With regard to this point, the following has been added in the relevant part as the limitation of this research. Thank you for your valuable feedback. We would like to incorporate your opinions into our Future work.

Page 26, Lines 344-349: This study did not investigate whether microorganisms present in the meat are the same as those identified on the equipment and surfaces. To clarify this, strain identification using molecular typing methods, such as RAPD, MLST, and ribotyping, need to be performed [24-29]. Further detailed research should be carried out in the future to determine what kinds of instruments and bacterial species are likely to contaminate the surface of carcasses.

---

## [Editor Report · Decision Letter 2]

29 Apr 2022

Distribution of psychrophilic microorganisms in a beef slaughterhouse in Japan after cleaning

PONE-D-22-01666R2

Dear Dr. Takahashi,

We’re pleased to inform you that your manuscript has been judged scientifically suitable for publication and will be formally accepted for publication once it meets all outstanding technical requirements.

Kind regards,

Guadalupe Virginia Nevárez-Moorillón, Ph.D.

Academic Editor

PLOS ONE
---

## [Editor Report · Acceptance letter]

8 Jul 2022

PONE-D-22-01666R2 

Distribution of psychrophilic microorganisms in a beef slaughterhouse in Japan after cleaning 

Dear Dr. Takahashi:

I'm pleased to inform you that your manuscript has been deemed suitable for publication in PLOS ONE. Congratulations! Your manuscript is now with our production department. 

Kind regards, 

on behalf of

Dr. Guadalupe Virginia Nevárez-Moorillón 

Academic Editor

PLOS ONE